# What factors shape an individual's probability to be enrolled in a professionally-managed community-based health insurance? Results from a cross-sectional case-control study in two districts in Mali

Hamidou Niangaly[1]*, Manuella De Allegri[2], Laurence Touré[3], Dansiné Diarra[4], Valéry Ridde[5,6,7]

1 Institut National de Santé Publique, Bamako, Mali, 2 Heidelberg Institute of Global Health, Heidelberg University Hospital and Faculty of Medicine, Heidelberg University, Germany, 3 Miseli, Bamako, Mali, 4 Université des Sciences Sociales et de Gestion de Bamako, Bamako, Mali, 5 Institut de Santé et Développement, Université Cheikh Anta Diop, Dakar, Sénégal, 6 Université Paris Cité, IRD, Inserm, Ceped, Paris, France, 7 Université Sorbonne Paris Nord, Paris, France

* hamidouniangaly@gmail.com

## Abstract

Across West Africa, countries are promoting new models of community-based health insurance (CBHI) to overcome the challenges faced by existing schemes. We investigated factors associated with an individual's probability to be enrolled in one of two professionally-managed CBHI in Mali. We carried out a case-control methodology and used multi-stage sampling to select CBHI members and non-members. The primary outcome variable was being enrolled or not in the CBHI. Three-level mixed-effects logistic regression model was used to estimate the association between individual enrolment and series of individual, household, and wider community factors. Of the 1,270 people surveyed 847 and 423 were non-members and members respectively. Respondents with primary education (AOR = 1.6, P = 0,035), secondary education (AOR = 2.31, P = 0,001) or higher school/University (AOR = 5.5, P < 0,001); who were economically active (AOR = 1.52, P = 0,05) and considered themselves wealthy (AOR = 1.94, P < 0,001) were more likely to subscribe to CBHI. Those who reported having a good perception of their health (AOR = 0.68, P = 0,038) were less likely to subscribe to CBHI. Albeit substantially higher than what experienced by community-based schemes, membership in these professionally-managed CBHI remains low and the determinants of participation do not differ substantially from what reported in the general literature. Targeting the needs of poor vulnerable people and more pervasive communication strategies are urgently needed to enhance enrolment in these new CBHI and ensure the way to UHC.

**Data availability statement:** All relevant data are within the paper.

**Funding:** This work was supported by the Agence Française de Développement (AFD). This research is part of the Unissahel program (Universal Health Coverage in Sahel), funded by the AFD. The funders had no role in study design, data collection and analysis, decision to publish, or preparation of the manuscript. The opinions expressed are exclusive of the authors and do not reflect the official position of the AFD.

**Competing interests:** The authors have declared that no competing interests exist.

## Introduction

Community-based health insurance (CBHI) operates voluntarily, is managed by volunteers, and is characterized by community members pooling funds to offset healthcare costs [1]. In West Africa, CBHI schemes, usually referred to as *Mutuelles*, are traditionally set up on a small scale, so that risk sharing is confined to the municipality level [2]. For many years, research on health financing reforms in several African countries has shown that relying on local CBHI for universal health coverage (UHC) is ineffective. In French-speaking West Africa, CBHIs are often called "mutuelles de santé". They are local micro-insurance schemes based on solidarity, autonomy and participatory democracy. Membership is voluntary and governance is community-based. Since the 1980s, the CBHI have been established across West Africa with the help of technical and financial partners, such as World Bank and U.S. Agency for International Development [3]. However, despite this external support, these CBHI consistently suffered from low penetration rates and never achieved high population coverage [4]. Many studies carried out in the 1990s and 2000s tried to understand factors associated with population acceptance to support projects and States investing in CBHI as a temporary instrument towards fostering Universal Health Coverage (UHC) [4–7]. The main factors identified to explain low participation in small-scale volunteer-run *CBHI* are: low contributory capacity of villagers, lack of trust in the management structures, poor perceptions of quality of care provided, limited provision of services, the persistence of co-payments and the periodicity of contributions, previous negative experiences with community associations [6–9].

In the early 2020s, in light of persisting low penetration rates and in light of the criticism by the World Health Organization (WHO), clearly pointing out at the system weaknesses, some countries in the region returned to a suggestion once advanced by Bart Criel [10] in the 1990s and proposed to expand the *CBHI* risk pooling at the district level. For instance, Mali introduced two pilot *CBHI* managed at district level and by professionals [11,12]. Details of these schemes' implementation have been described before [11,12]. In 2018, less than 3% of Mali's population was covered by a CBHI [11], but these two *CBHI* alone managed to cover 9% of the population in less than three years of operation. While this success is noteworthy, it remains relatively low considering overall coverage needs.

To support Malian officials looking for solutions to expand UHC, it is important to understand what factors contributed to people's decision to join this new type of *CBHI*. To our knowledge, no other study in West Africa has explored determinants of participation in CBHI managed by professionals and organized at a district level, even in Senegal where they have been in place in two departments since 2014 [13]. This is important since it represents a fundamental departure from the traditional CBHI management model, requiring people to trust a much larger structure.

We aimed to fill this gap in knowledge by examining what factors shape participation to district and professionally-run *CBHI* in Mali. Our ultimate objective is ultimately to inform public officials in their decision-making towards UHC.

PLOS Global Public Health

## Materials and methods

### Study sites

As part of the Health and Social Development Support Program (PADSS2) in the Mopti Region, two professionally managed *CBHI* have been implemented in the administrative circles (districts) of Mopti and Bandiagara (Fig 1). The circles of Mopti and Bandiagara are located in central Mali, in the Mopti region, with the populations of 492 970 and 418 873 inhabitants in 2018 respectively according to estimates by the National Population Direction. The Mopti region with a population of 3 209 999 inhabitants in 2024, is Mali's fifth administrative region. Located in the center of the country, it covers an area of 79,017 km2, the security crisis that began in Mali in 2012 spread to the Mopti region in 2015 and worsened significantly in 2019. The intensification of violence, including attacks by armed groups and intercommunal conflicts, has led to massive population displacements as communities seek safety in more secure areas. This instability has also severely affected the healthcare system, as many healthcare professionals have been forced to flee, reducing access to essential medical services.

### Study period

The study was carried out over 6 months, including the preparatory phase, data collection and report writing. Data collection took place from June 2 to August 8, 2022. We conducted the study during this period due to the availability of resources required to carry out the activities. Data collection took place during the months of July and August, which coincided with the start of the agricultural season. As a result, the surveyors had to revisit several households in order to conduct interviews with the selected members of the community-based health insurance (CBHI) schemes. These

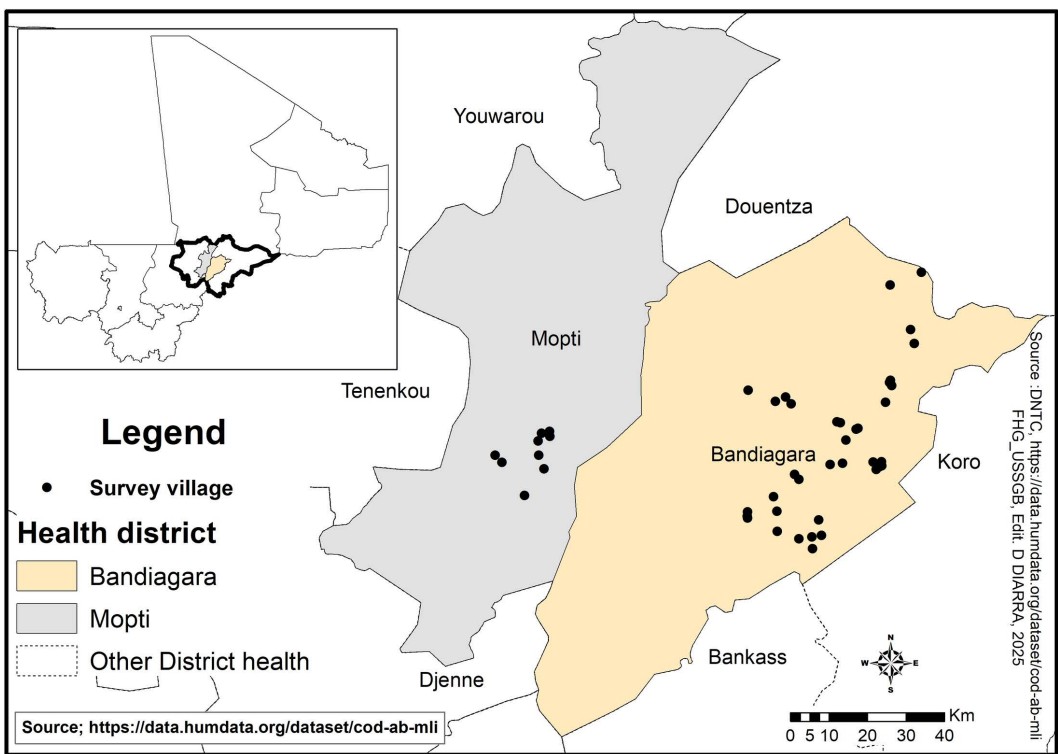

**Fig 1. Study sites.** Link to the base layer of the map: https://data.humdata.org/dataset/cod-ab-mli. Link to the terms of use/ license information for shapefile: https://data.humdata.org/dataset/cod-ab-mli.

interviews were conducted at the time that was most convenient for the participants. Appointments were scheduled in advance, either by phone or through their relatives. Most of the interviews were conducted in the afternoon or evening. These efforts together ultimately allowed us to ensure that the expected sample could be achieved. However, we had to replace some of the initially selected members of the community-based health insurance schemes (CBHI), as interviews could not be conducted with them despite our efforts. This may have introduced a selection bias, potentially limiting the generalizability of the findings to the broader target population. Nevertheless, since these members represented about 2% of the total sample, the impact of this bias on the overall results is likely to be marginal.

### Study population

The study population was the target population of *CBHI* which represents 78% of the population of the Mopti and Bandiagara circles. This population is made up of workers in the informal sector which is neither covered by the compulsory health insurance reserved for employees and civil servants nor by the medical assistance scheme intended solely for indigent persons as defined by the social services administration. We restricted our study to persons aged 18 and older.

### Management of the CBHI

The two community-based health insurances are managed by agents trained in the implementation of a CBHI system through the technical support of the "Union technique de la mutualité malienne" and receive financial support from the "Programme de santé et de développement social". Elected CBHI representatives were elected in each **municipality** to better communicate about this innovation in the villages. Employee managers were recruited according to the size of the municipality (one manager for two or three municipalities). Their role was to communicate about **CBHI** and their products, but also to monitor membership and collect contributions. This professional management thus makes these **CBHI** different from those experienced in previous research in Mali and elsewhere [13–15]. Each CBHI was managed by a team composed of a coordinator, a medical advisor, development assistants, managers and financial management agents. The coordinators were specialized in social protection management or community health system management. The medical advisors were physicians. The development assistants and managers held at least a bachelor's degree in social sciences, finance or administration. And the financial agents held Bachelor or Master degrees. All of these professionals were recruited and trained by the Union Technique de la Mutualité Malienne (UTM), the umbrella organization for mutual health schemes in Mali. In addition to human resources, material resources—such as IT equipment and logistical tools—were provided to ensure the professional management of the schemes.

### Sampling

We relied on a three-stage random sampling to select municipalities, villages, and members of the *CBHI* in each circle (Fig 2). As described hereafter, we combined random and purposive sampling technique to select the communities where to conduct our survey. Security considerations imposed this choice. We defined as cases individual who had paid their full membership fees and for whom the insurance coverage had taken effect, regardless of the status of their contribution (updated or not). We defined as controls all individuals without a CBHI membership.

### Sample sizes and selection procedure

**Municipalities.** The municipality is an administrative division that groups together several villages. Due to insecurity, we first drew a list of accessible areas in agreement with village, political, and local administrative authorities. Out of 15 municipalities in the circle of Mopti, three municipalities were accessible (20%): Socoura, Mopti, and Sio. In the Bandiagara circle, 10 out of 21 municipalities were accessible (47.6%): Sangha, Kendié, Wadouba, Soroly, Bandiagara, Dandoli, Dourou, Diamnati, Pelou, and Ségué Iré. The survey was carried out in the three accessible municipalities of Mopti and the 10 accessible municipalities of Bandiagara, i.e., on average 36.1% of the municipalities of the two circles.

**Global Public Health** PLOS

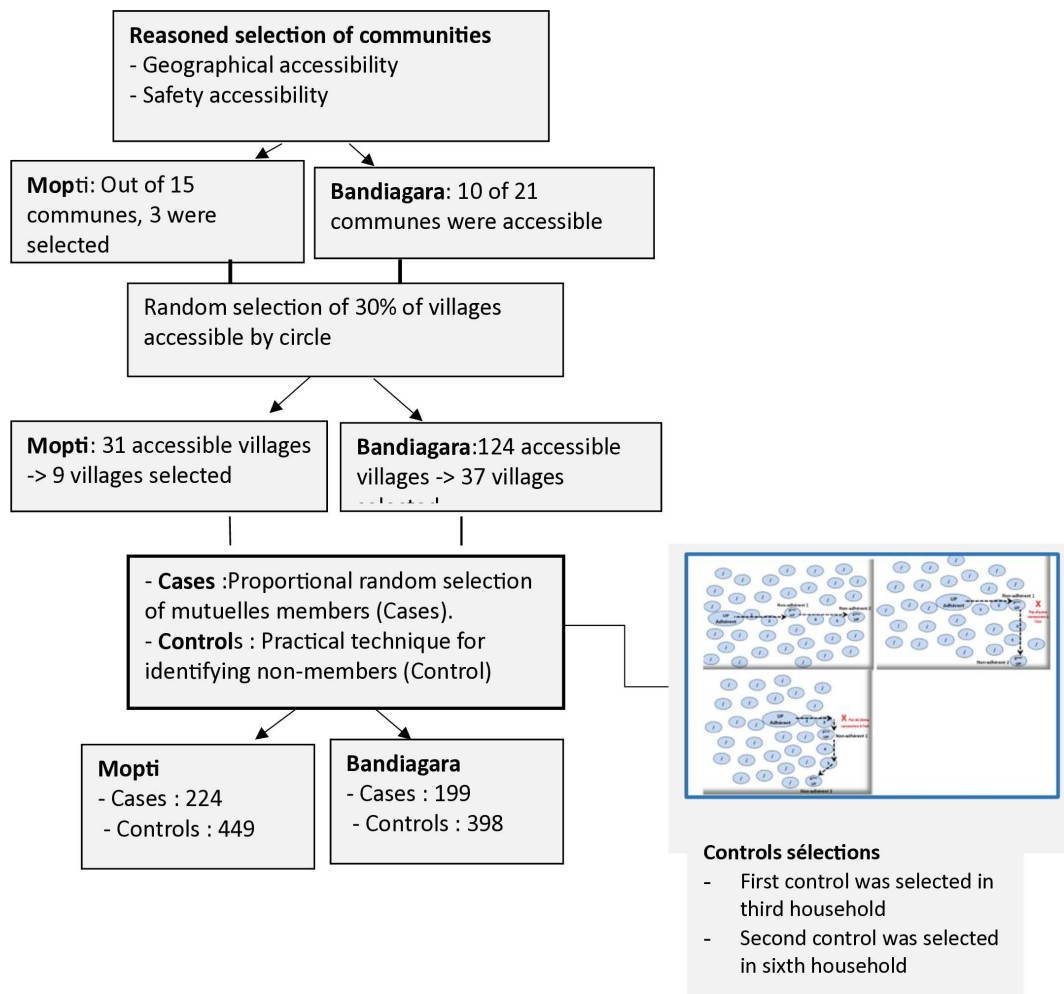

**Fig 2. Flow chart.**

**Villages.** From all villages included in the accessible municipalities, we further excluded villages with five or fewer members, due to feasibility concerns. This resulted in a total sampling frame of 155 villages, 124 and 31 in Bandiagara and Mopti respectively. From this total of 155 villages, we selected 46 villages, 37 in Bandiagara and 9 in Mopti (sampling proportional to circle size) to be included in our survey.

The inability to include members from localities located in insecure areas in the survey represents a key limitation of this study, as it may hinder the generalizability of the results to the broader target population.

**Members and non-members.** To identify respondents to include in our study, we relied on the single population proportion formula [16] and adopted the proportion of two controls (non-*CBHI* members) to one case (*CBHI* member). The formula below (Equation 1) was used to calculate the sample size for each group (cases and controls):

$$n = z^2 * \frac{pq}{d^2}$$

(1)

Where p = 1-q, *z* is the value dependent on the chosen error risk α (z = 1.96 for α = 5%); d, the desired precision, and n = size of the sample. For an average penetration rate of the *CBHI* in the Mopti and Bandiagara circles in December 2021 of 9.1% (n/N = 12619/138670) which corresponds to the value q, 1-q = 0.91, a desired precision of 3%, a confidence level

of 95%, a design effect of 1.5 and a non-response rate of 10%, a total size of 1270, i.e., 423 members and 847 non-members was determined to be sufficient to identify factors associated with membership in the *CBHI*.

**Recruitment of members (cases).** We aimed at producing a sample of *CBHI* members proportional to number of members in each village. We could rely on official membership lists to guide our sampling and drew our samples at random from these lists. However, due to security reasons, we could not always locate the households we had identified and had to adjust sampling to the reality of the fact that some members had relocated due to insecurity concerns.

**Recruitment of non-members (Controls).** Given the absence of adequate census data from which to draw the control sample, we had to rely on pragmatic techniques to identify controls. Hence, as first control, we selected the third compound to the East of the *CBHI* member respondent and as second control, we selected the third compound to the East of the latter. In case non suitable compound could be identified in the Eastward direction, the investigator turned South to identify an appropriate control compound.

Once in a compound, controls were selected at the individual level to match the CBHI member case's sex and status within the household (i.e., household head, spouse, son, or daughter). Moreover, in the case of multi-household compound, households were selected to match the same household status as the corresponding member household (e.g., dominant versus subordinated household).

## Variables and their measurement

### Dependent variable

The dependent variable was defined as membership in the CBHI. It was specified as equal to 1 if the person was enrolled, and as equal to 0 if not. As mentioned earlier, membership was defined as having joined the scheme at any one point in time, with no actual checks of whether the member was up to date with the payment of the membership fee. Members will have all started to benefit from insurance coverage.

### Independent variables

The independent variables selected were those previously found to be associated with membership in CBHI [17–20]. Independent variables at the individual level measured the socio-demographic profile of members and non-members. They included sex, age, level of education, participation in economic activity, membership status in a formal organization, wealth status, perceived health status, whether or not chronic illness was present, whether or not a disability was present, and the source of information on the existence of a *CBHI* in the locality. Independent variables at the household level included socio-economic status, derived combining household material assets using principal component analysis. Notably, the main component of the polychoric correlation matrix [21,22] was used to determine the household wealth index. Following the guidelines of the Demographic and Health Survey, wealth quintiles were created by ordering the wealth index into five equal parts [23]. Households were accordingly classified as very poor, poor, intermediate, less poor, and least poor. At village level, we differentiated urban and rural localities.

### Empirical approach

Given our binary dependent variable, we adopted a logistic regression model. Since individuals were nested in villages, the assumption of independence inherent of standard logistic regression was violated and we had to rely on a multi-level model, to take into account the hierarchical structure of our data and the resulting intra-cluster variability. Following Bekele 2022, we used a three-level mixed-effects logistic regression model to determine the factors that influence decision to join or not to join enrolment in the CBHI membership [15]. We modeled the probability of being registered in the CBHI using a three-level model, estimating the equations ((2) and (3)) below:

$$log\left(\frac{\pi_{ij}}{1-\pi_{ij}}\right) = \beta_0 + \beta_1 X_{ij} + \beta_2 F_{ij} + \mu_j$$

(2)

PLOS Global Public Health

$$\mu_j \sim \left(\sigma_\mu^2\right) \tag{3}$$

Where i and j are the units of level 1 (individuals = survey respondents) and level 3 (village and municipality), respectively; X and F apply to the variables of the level of the individual and the municipality and village respectively, $\pi_{ij}$ is the probability of enlisting in the CBHI in the community for the third individual; β are the fixed coefficients-therefore, there is a corresponding efficiency for each increase of a unit of X|F (a set of predictors). Whereas in the absence of control of predictors, β0 is the intercept, capturing the effect on the probability of the individual to subscribe to the CBHI; and $\mu_j$ indicates the random effect for the municipality and village (effect of the municipality or village on the individual's decision to subscribe to the CBHI). The existence of cluster data and variations within and between communities (municipality and village) were taken into account assuming that each community had a different intercept (β0) and fixed coefficient (β). The log-likelihood ratio (LLR), Akaike Information Criteria (AIC) and Schwartz-Bayesian Information Criteria (BIC) tests were used to evaluate the best fit model [24]. The Wald test was used to investigate the random effects at village and municipality- levels [25]. The importance of incorporating random parameter estimates was investigated using intraclass correlation test [24,26]. The near-zero intraclass correlation test (ICC) values indicate that the characteristics of persons surveyed within municipalities or villages are no more similar than those of persons surveyed across different municipalities or villages. More details on tests are given in supporting information section. Data have been analyzed using Stata software (15) (Stata Statistical Software: Release 14. College Station, TX: StataCorp LP). The *melogit* code has been used to fit mixed logit model. Appropriate statistical tests (Student, Pearson) were used to measure associations with an alpha risk of 5%.

## Ethical clearance

This study is part of an extensive research program (Couverture Universelle Santé au SAHEL), which was approved by the institutional ethics committee of the National Institute of Public Health of Mali, on December 26, 2018, (decision no. 34/2018/CE-INRSP) and authorized by the Minister of Health and Social Affairs in 2020 (authorization number: 001744 MSAS-SG). Verbal consent was obtained from all participants before data collection. We asked permission from community leaders and local authorities. Before the start of data collection, the delegates from the community-based health insurance scheme informed community leaders and local government representatives of the research. We sought community permission from customary leaders such as village chiefs. In each village, the investigators visited the village chief to explain the purpose and method of the research. The authorization was obtained from local health officials and government representatives before the survey.

## Individual consent

This survey included only individuals who were at least 18 years old at the time of the study. Individual informed verbal consent was systematically obtained before enrolling participants.

When a volunteer meeting the inclusion criteria was identified, the study's objectives and procedures were clearly explained to them in a language they understood. For illiterate participants, a witness proficient in French and capable of translating the consent content into the participant's language was involved to ensure full comprehension.

The interview only began after the volunteer had given their consent. It was clearly stated that they could withdraw from the survey at any time or request the deletion of their information without any consequences. If the participant had any questions or concerns, they could contact the principal investigator or the ethics committee, whose phone numbers were provided.

The questionnaire included a specific question to confirm the participant's consent: "Do you voluntarily agree to participate in the study?" If the answer was NO, the survey was immediately terminated. Only volunteers who explicitly agreed to participate were interviewed.

## Protocol deviation

With the exception from the change in the process of obtaining individual consent, the protocol was implemented in accordance with its initial plan. In the protocol approved by the ethics committee, written consent was initially required. However, due to the evolving security situation in the field and following the advice of local community leaders, researchers opted for verbal consent. This modification in protocol implementation was submitted to the ethics committee and received its approval. The ethics committee's approval letter is attached as an appendix to the manuscript.

## Inclusivity in global research

The supporting information (S1 Text) includes additional information regarding the ethical, cultural, and scientific considerations specific to inclusivity in global research.

## Results

### Characteristics of the persons surveyed

Table 1 provides a summary of the characteristics of members and non-members. A total of 1,270 people were surveyed, including 847 non-members and 423 members. There were more men than women among both members and non-members, with 70.8% (600/847) and 72% (303/423) respectively. Large proportions of non-members (47.7%) and members (32.4%) were out of school, P<0.001. Among those who attended school, with the exception of primary level, where there was no statistically significant difference between the two comparison groups, the proportion of respondents who had reached secondary school and university levels was higher among members than non-members, P<0.05. Regarding participation in economic activities, there were more economically active people among members (86.5%) than non-members (78%), P=0.001. Respondents who were members of a formal organization were more likely to be members (24.8%) of a CBHI than non-members (13.7%), P=0.001. Those who considered themselves wealthy were also more numerous among members (25.3%) than non-members (11.8%), P=0.001. The percentage of respondents with chronic illness was higher among members (17%) than among non-members (12.3%). Regarding the household wealth quintile, 47.5% of members and 36.3% of non-members were in less/least poor households, P=0.001. The two groups were statistically comparable in terms of disability, those who considered themselves in good health and sources of information (radio, television, village authorities) and location (rural or urban)

### Factors associated with subscription to new CBHI

**Predictors at the individual level.** Table 2 presents estimates using the three-tiered mixed-effects logistic regression method. Columns (1), (2), and (3) contain Crude Odds Ratios (CORs), confidence interval values (CI 95%), and probability values (P-value). Columns (4), (5), and (6) contain Adjusted Odds Ratios (AORs), confidence interval values (CI 95%), and probability values (P-value). Respondents' level of education, participation in economic activity, perceived wealth, perception of good health, and household size were significant predictors of new CBHI. Compared to respondents who were not in school, those who had reached primary level (between 1 and 6 years of education), second cycle level (between 7 and 9 years of study) and high school or university level (10 years of study or more) were respectively 1.6 times (AOR=1.6; CI 95%=1.03-2.48; P=0.035), 2.31 times (AOR=2.31; CI å95%=1.43-3.71; P=0.001) and 5.5 times (AOR=5.5; CI 95%=3.31-9.14; P<0.001) more likely to subscribe to the CBHI. Economically active respondents or those who considered themselves wealthy were 1.52 times (AOR=1.52; CI 95%=1-2.31; P=0.05) and 1.94 times (AOR=1.94; CI 95%=1.35-2.79; P<0.001) more likely to subscribe to the CBHI than those who had no activity or those who declared to be poor. Regarding perceived health status, respondents who reported being in good health were 0.68 times (AOR=0.68; CI 95%=0.47-0.98; P=0.038) less likely to subscribe to the CBHI than those who felt in poor health.

**Table 1. Characteristics of survey participants.**

| Variables | Categories | No-member | | Member | | P-value |
|---|---|---|---|---|---|---|
| | | Freq. | % | Freq. | % | |
| Gender | Female | 247 | 29,2 | 120 | 28,4 | 0,77 |
| | Male | 600 | 70,8 | 303 | 71,6 | |
| Education level | Education No | 404 | 47,7 | 137 | 32,4 | < 0.001 |
| | Primary | 111 | 13,1 | 54 | 12,8 | |
| | Secondary | 72 | 8,5 | 55 | 13 | |
| | College/University | 59 | 7 | 93 | 22 | |
| | Koranic | 201 | 23,7 | 84 | 19,9 | |
| Economically active | No | 186 | 22 | 57 | 13,5 | < 0.001 |
| | Yes | 661 | 78 | 365 | 86,5 | |
| Member of formal organization | No | 731 | 86,3 | 318 | 75,2 | < 0.001 |
| | Yes | 116 | 13,7 | 105 | 24,8 | |
| Consider yourself rich | No | 747 | 88,2 | 316 | 74,7 | < 0.001 |
| | Yes | 100 | 11,8 | 107 | 25,3 | |
| Has disability | No | 816 | 96,3 | 406 | 96 | 0.752 |
| | Yes | 31 | 3.7 | 17 | 4 | |
| Has chronic disease | No | 743 | 87,7 | 351 | 83 | 0,021 |
| | Yes | 104 | 12,3 | 72 | 17 | |
| Self-reported health status | Bad | 191 | 22,6 | 106 | 25,1 | 0,319 |
| | Good | 656 | 77,4 | 317 | 74,9 | |
| Household wealth | Poorest | 187 | 22,1 | 67 | 15,8 | < 0.001 |
| | Poor | 199 | 23,4 | 57 | 13,5 | |
| | Middle | 154 | 18,2 | 98 | 23,2 | |
| | Less poor | 164 | 19,4 | 90 | 21,3 | |
| | Least poor | 143 | 16,9 | 111 | 26,2 | |
| Source of information about the CBHI | | | | | | |
| Radio | No | 388 | 70,8 | 309 | 73 | 0,44 |
| | Yes | 160 | 29,2 | 114 | 27 | |
| TV | No | 499 | 91,1 | 372 | 87,9 | 0,11 |
| | Yes | 49 | 8,9 | 51 | 12,1 | |
| Village authorities | No | 487 | 88,9 | 371 | 87,7 | 0,58 |
| | Yes | 61 | 11,1 | 52 | 12,3 | |
| Location | Rural | 553 | 65,3 | 276 | 65,2 | 0,99 |
| | Urban | 294 | 34,7 | 147 | 34,8 | |

Source: Authors

The probability of subscription to the CBHI was 1.07 times higher among households with a large number of people than those with few people (AOR = 1.07; CI 95% = 1.02-1,11; P = 0.003).

**Predictors at the Community level**

The only variable that characterized the community level in our study was the level of development of the villages where the respondents lived (rural or urban). Respondents who lived in rural areas had the same subscription level to the CBHI as those living in urban areas (AOR = 1.04; CI 95% = 0.75-1.46, P = 0.8).

**Table 2. Factors associated with enrollment in the CBHI.**

| Variables | Categories | (1) COR | (2) CI95% | (3) P-value | (4) AOR | (5) CI95% | (6) P-value |
|---|---|---|---|---|---|---|---|
| Gender | Male | Ref. | – | – | Ref. | – | – |
| | Female | 1,04 | 0,80-1,35 | 0,769 | 0,83 | 0,59-1,18 | 0,299 |
| Age | | 0,99 | 0,99-1,00 | 0,139 | 1 | 0,98-1,01 | 0,588 |
| Education level | Education No | Ref. | – | – | Ref. | – | – |
| | Primary | 1,43 | 0,98-2,09 | 0,062 | 1,60 | 1,03-2,48 | 0,035 |
| | Secondary | 2,25 | 1,51-3,36 | < 0.001 | 2,31 | 1,43-3,71 | 0,001 |
| | College/University | 4,65 | 3,18-6,79 | < 0.001 | 5,50 | 3,31-9,14 | < 0.001 |
| | Koranic | 1,23 | 0,89-1,70 | 0,201 | 1,36 | 0,93-1,98 | 0,11 |
| Economically active | No | Ref. | – | – | Ref. | – | – |
| | Yes | 1,80 | 1,30-2,49 | < 0.001 | 1,52 | 1,00-2,31 | 0,05 |
| Member of formal organization | No | Ref. | – | – | Ref. | – | – |
| | Yes | 2,08 | 1,55-2,79 | < 0.001 | 1,3 | 0,92-1,84 | 0,142 |
| Consider yourself rich | No | Ref. | – | – | Ref. | – | – |
| | Yes | 2,53 | 1,87-3,42 | < 0.001 | 1,94 | 1,35-2,79 | < 0.001 |
| Has chronic disease | No | Ref. | – | – | Ref. | – | – |
| | Yes | 1,47 | 1,06-2,03 | 0,022 | 1,43 | 0,93-2,19 | 0,1 |
| Has disability | No | Ref. | – | – | Ref. | – | – |
| | Yes | 1,1 | 0,60-2,02 | 0,752 | 1,18 | 0,58-2,40 | 0,652 |
| Self-reported health status | Bad | Ref. | – | – | Ref. | – | – |
| | Good | 0,87 | 0,66-1,14 | 0,32 | 0,68 | 0,47-0,98 | 0,038 |
| Household size | | 1,06 | 1,03-1,09 | < 0.001 | 1,07 | 1,02-1,11 | 0,003 |
| Household wealth quintile | Poorest | Ref. | – | – | Ref. | – | – |
| | Poor | 0,8 | 0,53-1,20 | 0,28 | 1,03 | 0,65-1,64 | 0,899 |
| | Middle | 1,78 | 1,22-2,59 | 0,003 | 1,4 | 0,90-2,20 | 0,138 |
| | Less poor | 1,53 | 1,05-2,24 | 0,028 | 1,35 | 0,87-2,12 | 0,185 |
| | Least poor | 2,17 | 1,49-3,15 | < 0.001 | 1,49 | 0,94-2,36 | 0,086 |
| Source of information | | | | | | | |
| Radio | No | Ref. | – | – | Ref. | – | – |
| | Yes | 0,89 | 0,67-1,19 | 0,441 | 0,75 | 0,54-1,04 | 0,085 |
| TV | No | Ref. | – | – | Ref. | – | – |
| | Yes | 1,4 | 0,92-2,11 | 0,115 | 1,33 | 0,82-2,15 | 0,253 |
| Village authorities | No | Ref. | – | – | Ref. | – | – |
| | Yes | 1,12 | 0,75-1,66 | 0,576 | 1,33 | 0,86-2,08 | 0,202 |
| Location | Rural | Ref. | – | – | Ref. | – | – |
| | Urban | 1 | 0,78-1,28 | 0,988 | 1,04 | 0,75-1,46 | 0,8 |

**Random intercepts**

| | | | | | Variance | SE | IC95% |
|---|---|---|---|---|---|---|---|
| Municipality | | | | | 3.44E-38 | 2,85E-21 | — |
| Village | | | | | 1.50E-34 | 2,21E-18 | — |
| Number of observations | | | | | 970 | | |

**Model statistics -values**

| | | | | | | | |
|---|---|---|---|---|---|---|---|
| Chi 2 | | | | | 106,68 | | |
| BIC | | | | | 1352,89 | | |
| AIC | | | | | 1245,59 | | |
| Log Likelihood | | | | | —600,79 | | |

*(Continued)*

**Table 2.** (Continued)

| Variables | Categories | (1) COR | (2) CI95% | (3) P-value | (4) AOR | (5) CI95% | (6) P-value |
|---|---|---|---|---|---|---|---|
| Intraclass correlation (ICC) | | | | | ICC | SE | IC95% |
| Municipality | | | | | 1.05E-38 | 8,67E-22 | — |
| Village | | | | | 4.56E-35 | 6,73E-19 | — |

Notes: Columns (1), (2) and (3) contain the Crude Odds Ratio (COR), Interval confidence at 95% (IC95%) and the values of the probability (P-value). Columns (4), (5), and (6) contain the adjusted odds ratio (AOR), Interval confidence at 95% (IC95%), and the values of the probability (P-value); Chi 2 is the model fitness test under the Chi-squared distribution, BIC = Shwartz-Bayesian Information Criteria, AIC = Akaike Information Criteria, ICC = Intra Class Correlation. **Source: Authors**

### Random effects variation measures

Estimates of random effects at the municipality (Variance = $3.44 \times 10^{-38}$; SE = $2.85\ 10^{-21}$) and village (Variance = $1.5 \times 10^{-34}$; SE = $2,21 \times 10^{-18}$) levels show very low variances. This suggests that municipalities and villages were similar. None of the model fit tests (AIC, BIC, LLR) discriminates between the different models. The Wald test statistics at the municipality (Wald statistics = $1.47 \times 10^{-34}$, P > 0.05) and the village (Wald statistics = $4.6 \times 10^{-33}$, P > 0.05) levels were not statistically different from zero. Based on the results of the Wald test, we cannot conclude to the presence of random effects at the supra-individual level (municipalities and villages) that influence the decision to enter in the CBHI. However, given the hierarchical structure of our data, the three-tiered mixed effects logistics model was used as appropriate to determine predictive factors for CBHI subscription.

## Discussion

This study makes a unique contribution to the literature by being one of the first studies examining determinants of participation to the new professionalized CBHI, within the context of the newly restructured CBHI system in two administrative circles (Mopti and Bandiagara) in Mali. Despite the professionalization of the CBHI, the beneficiary coverage rate remained low (9%) in the context of this project. This low membership level is reminiscent of the difficulty of CBHI in reaching the expected coverage of their target population [27]. Our study reveals that factors associated with participation in a professional CBHI do not differ from those associated with a traditional volunteer-based CBHI [8,9,28–31]. Thus, despite the relative effectiveness of the project, which has been able, in less than three years, to reach a penetration rate never achieved before, the same set of socio-demographic and economic factors continue to explain who is covered and who is not covered. One of the intrinsic limitations of our quantitative study is that we are unable to better understand the role of cultural factors or trust in the explanatory factors [32]. These more social dimensions are better understood through qualitative research, and we hypothesise that the influence of culture does not really change with the professionalization of CBHI staff. However, trust is certainly an important factor, as has been noted in other research on CBHIs and in our own work in Mali [33–36]. In addition, trust in health professionals also has an important influence on the willingness to join CBHIs and thus to benefit from quality health service coverage [37,38]. But studies in West Africa have shown that the relationship between health professionals and member CBHIs is not always positive, with the result that non-members are sometimes favoured [38,39].

Factors capturing an individual socio-economic status (level of education, perception of one's wealth status and ability to pay; household size, and perceived health status) were all found to be significantly associated with one's probability of being enrolled in the CBHI.

Our findings on the effect of socio-economic status resonate with what has been observed in other settings, such as Mali, Senegal, Ghana and Asia countries [8,9,28–31]. This is not surprising since economically active and better-off

individuals have a higher purchasing power and, therefore, enjoy a higher ability to pay the scheme premiums. It is important to recall that the State of Mali, aware of the challenges of membership, subsidizes the rate of contributions to all CBHI in the country up to 50%, as in Senegal [40]. These two professionals CBHI received an additional 30% subsidy, funded by donors. This 80% subsidy of the real premium probably explains the relative success observed in terms of enrollment, reaching 9%, but leaves open questions in terms of the equity and sustainability of the subsidization, since it appears that poorer individuals were still less likely than least-poor individuals to enroll. It needs to be noted here that in principle, the ultra-poor (*indigents*) are entitled to receive services completely free of charge thanks to the RAMED policy [41,42]. Still, not only is policy implementation far from being effective, but the proportion of individuals covered is too small compared to the actual inability to join the scheme observed in our study. A similar situation is also observed in Senegal, where the *indigents* have a right to join the CBHI free of charge within the framework of a national exemption policy but do not do so due to a lack of information and resources [43,44]. The challenges of targeting the indigent and then implementing policies to help them are well-known in the region [42,45–47] but do not seem to have been considered by the designers of the intervention. While the national policy is highly original, simply waiving the CBHI membership fee is not enough. The indigent faces many other determinants of access to care and it would be essential to deploy more active support measures. Health navigation initiatives could be tested and the support of community health workers sought [48].

The relationship between higher educational and enrollment in CBHI has been observed before in settings such as Burkina Faso and Ethiopia countries [7,20,49]. This relationship is a constant in most studies in Africa on the factors that encourage people to join CBHI [31,50]. This can be explained by the fact that higher educational levels increase awareness of risk minimization strategies and hence decision to join health insurance schemes. The fact that we observe an effect of education, independently of socio-economic status, calls into question promotion activities, clearly pointing at the need for more targeted information and communication campaigns. Comparisons with other countries in French-speaking West Africa are tricky, as few countries have yet tried out professional CBHIs, with the exception of Senegal, which has had two departments since 2014. Furthermore, in Senegal, to our knowledge, no research has yet compared the membership factors between members and non-members of professional CBHIs. However, recent studies have shown the challenges of equity and membership for the poorest members of these CBHIs, even if their membership is fully subsidised by the state [44].

In line with prior studies, we also observed that individuals who perceived themselves to have a higher health status were less likely to join the CBHI [51]. This is likely to be an indication of adverse selection, and represents a risk for the long-term financial viability of the scheme. The challenge the CBHI face is to change risk perceptions among the target populations to attract a wider constituency of beneficiaries and by doing so, enhancing the financial viability of the scheme.

The implications for public health of this research are clear, even if they are not very original. Firstly, it confirms the challenges of trying to base a UHC policy on CBHI with voluntary adherence [27], even if the CBHI is professional. Large-scale public funding remains an urgent solution for the UHC [50]. Secondly, it also confirms the compelling need to subsidise membership in a context of scarce resources and a region living in a fragile security environment. Finally, if we want to be serious about our commitments to equity [52] and UHC, we must combine CBHI membership and healthcare user fees exemption with support and health navigation measures for the worstoff. There is still an urgent need to organize action research in this area.

## Conclusions

Despite the challenges of operating in a setting affected by insecurity, this study relied on a rigorous methodology to assess factors associated with adherence to the innovative CBHI model of the professional CBHI in Mali. Two years into their implementation, these CBHI reached 9% of the target population.

While our study was not set to assess the effectiveness of the project, it allowed us to investigate what challenges to membership persisted. Considering the persistence of important socio-economic and educational inequities in individual enrollment, group enrollment strategies (cooperatives, schools, villages; etc.) could be tested as a viable option to overcome existing inequities. Moreover, ensuring that the ultra-poor are included through the systematic integration of targeted exemption policies in emerging insurance schemes is fundamental. Last but surely not least, the urgency remains to ensure access to quality health care in a country and region that faces unprecedented security challenges.

## Supporting information

**S1 Table. Statistical tests for choosing the appropriate model.** The BIC and AIC criteria are mathematical methods for assessing the suitability of a model to the data from which they were generated [53]. The model with smallest values of the BIC or AIC tests is the one that adjusts the data better. The LL test is a measure of the suitability of a model; the higher the value, the better the model. The ICC expresses the proportion of total variance at the common or village level [54]. The greater the value of the ICC, the more the characteristics of the groups (municipalities, villages) are different, and therefore, the model with mixed effects is appropriate. The results of AIC, BIC and LL tests do not discriminate between different models.
(DOCX)

**S1 Text. Inclusivity in global research.**
(DOCX)

## Acknowledgments

Thank you to all participants in the survey. We also deeply thank the investigators who braved difficult conditions on the ground, including insecurity, to gather quality information.

**Institutional review board statement:** The Ethics Committee authorized this research by Decision No. 34/2018/ CE-INRSP of 26 December 2018. All individuals were informed of the ethical issues and were allowed to withdraw from the study at any time. They all agreed to participate through verbal informed consent.

**Informed consent statement:** Informed consent was obtained from all subjects involved in the study.

## Author contributions

**Conceptualization:** Hamidou Niangaly, Manuella De Allegri, Laurence Touré, Valéry Ridde.

**Data curation:** Hamidou Niangaly, Dansiné Diarra.

**Formal analysis:** Hamidou Niangaly, Manuella De Allegri, Laurence Touré, Valéry Ridde.

**Funding acquisition:** Valéry Ridde.

**Investigation:** Hamidou Niangaly, Manuella De Allegri, Laurence Touré, Valéry Ridde.

**Methodology:** Hamidou Niangaly, Manuella De Allegri, Laurence Touré, Dansiné Diarra, Valéry Ridde.

**Project administration:** Laurence Touré.

**Resources:** Laurence Touré, Valéry Ridde.

**Software:** Hamidou Niangaly.

**Supervision:** Hamidou Niangaly, Manuella De Allegri, Laurence Touré, Dansiné Diarra.

**Validation:** Manuella De Allegri, Laurence Touré.

**Visualization:** Manuella De Allegri, Laurence Touré, Valéry Ridde.

**Writing – original draft:** Hamidou Niangaly, Valéry Ridde.

**Writing – review & editing:** Manuella De Allegri, Laurence Touré, Dansiné Diarra, Valéry Ridde.

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
