## [Decision Letter · Decision Letter 0]

PGPH-D-24-02388

What factors shape an individual’s probability to be enrolled in a professionally-managed Mutuelle? Results from a cross-sectional case-control study in two districts in Mali

Dear Dr. Niangaly,

Thank you for submitting your manuscript to PLOS Global Public Health. After careful consideration, we feel that it has merit but does not fully meet PLOS Global Public Health’s publication criteria as it currently stands. Therefore, we invite you to submit a revised version of the manuscript that addresses the points raised during the review process.

We look forward to receiving your revised manuscript.

Kind regards,

Ama Pokuaa Fenny, Ph.D

Academic Editor

Journal Requirements:

1. In the ethics statement in the Methods, you have specified that verbal consent was obtained. Please provide additional details regarding how this consent was documented and witnessed, and state whether this was approved by the IRB

2. Please include a complete copy of PLOS’ questionnaire on inclusivity in global research in your revised manuscript. Our policy for research in this area aims to improve transparency in the reporting of research performed outside of researchers’ own country or community. The policy applies to researchers who have travelled to a different country to conduct research, research with Indigenous populations or their lands, and research on cultural artefacts. The questionnaire can also be requested at the journal’s discretion for any other submissions, even if these conditions are not met. Please find more information on the policy and a link to download a blank copy of the questionnaire here: https://journals.plos.org/globalpublichealth/s/best-practices-in-research-reporting. Please upload a completed version of your questionnaire as Supporting Information when you resubmit your manuscript.

3. Please provide an Author Summary. This should appear in your manuscript between the Abstract (if applicable) and the Introduction, and should be 150–200 words long. The aim should be to make your findings accessible to a wide audience that includes both scientists and non-scientists. Sample summaries can be found on our website under Submission Guidelines:

https://journals.plos.org/globalpublichealth/s/submission-guidelines#loc-parts-of-a-submission.

4. In the online submission form, you indicated that "Data will be available at request.". 

a. In a public repository, 

b. Within the manuscript itself, or 

c. Uploaded as supplementary information.

Additional Editor Comments (if provided):

Reviewers' comments:

Reviewer's Responses to Questions

**Comments to the Author**

1. Does this manuscript meet PLOS Global Public Health’s publication criteria ? Is the manuscript technically sound, and do the data support the conclusions? The manuscript must describe methodologically and ethically rigorous research with conclusions that are appropriately drawn based on the data presented.

Reviewer #1: Yes

Reviewer #2: Partly

2. Has the statistical analysis been performed appropriately and rigorously?

Reviewer #1: I don't know

Reviewer #2: Yes

3. Have the authors made all data underlying the findings in their manuscript fully available (please refer to the Data Availability Statement at the start of the manuscript PDF file)?

Reviewer #1: Yes

Reviewer #2: Yes

4. Is the manuscript presented in an intelligible fashion and written in standard English?

Reviewer #1: Yes

Reviewer #2: Yes

5. Review Comments to the Author

Reviewer #1: The majuscriot is interesting and novel.

The comments for improvemnts has been added to the manuscript file attached.

I feel the manuscript lacks clear understanding at some places for the readers, please address that.

Reviewer #2: The study investigates factors influencing enrollment in professionally-managed community-based health insurance schemes (Mutuelles) in two districts in Mali. Using a case-control methodology, the research identified key predictors of enrollment, including higher education levels, economic activity, and perceived wealth. Those with better health perceptions were less likely to enroll, suggesting adverse selection. Despite professionalization efforts, coverage remains low, with only 9% penetration, highlighting the need for targeted communication strategies and equitable policies to increase membership and ensure progress toward universal health coverage (UHC).

The areas of improvement include:

Discussion of Equity Measures: While the article highlights the importance of equitable enrollment, it could delve deeper into strategies for reaching vulnerable populations.

Clarity in Statistical Analysis: Simplifying statistical explanations, particularly random effects and model selection, could enhance accessibility for a broader audience.

Policy Implications: The recommendations could be more actionable, outlining specific steps for policymakers to address barriers to enrollment.

Cultural Context: The role of cultural perceptions and trust in professional management could be better explored.

Comparative Analysis: Including comparisons to similar schemes in other regions could strengthen the findings' relevance and applicability.

6. PLOS authors have the option to publish the peer review history of their article (what does this mean? ). If published, this will include your full peer review and any attached files.

**Do you want your identity to be public for this peer review?** For information about this choice, including consent withdrawal, please see our Privacy Policy .

Reviewer #1: **Yes: ** SANJANA AGRAWAL

Reviewer #2: No

---

## [Decision Letter · Decision Letter 1]

PGPH-D-24-02388R1

What factors shape an individual’s probability to be enrolled in a professionally-managed Mutuelle? Results from a cross-sectional case-control study in two districts in Mali

Dear Dr. Niangaly,

Thank you for submitting your manuscript to PLOS Global Public Health. After careful consideration, we feel that it has merit but does not fully meet PLOS Global Public Health’s publication criteria as it currently stands. Therefore, we invite you to submit a revised version of the manuscript that addresses the points raised during the review process.

We look forward to receiving your revised manuscript.

Kind regards,

Ama Pokuaa Fenny, Ph.D

Academic Editor

Journal Requirements:

Additional Editor Comments (if provided):

Reviewer 4 has some comments that need to be addressed.

Reviewers' comments:

Reviewer's Responses to Questions

**Comments to the Author**

1. If the authors have adequately addressed your comments raised in a previous round of review and you feel that this manuscript is now acceptable for publication, you may indicate that here to bypass the “Comments to the Author” section, enter your conflict of interest statement in the “Confidential to Editor” section, and submit your "Accept" recommendation.

Reviewer #3: All comments have been addressed

Reviewer #4: All comments have been addressed

2. Does this manuscript meet PLOS Global Public Health’s publication criteria ? Is the manuscript technically sound, and do the data support the conclusions? The manuscript must describe methodologically and ethically rigorous research with conclusions that are appropriately drawn based on the data presented.

Reviewer #3: Yes

Reviewer #4: Yes

3. Has the statistical analysis been performed appropriately and rigorously?

Reviewer #3: Yes

Reviewer #4: Yes

4. Have the authors made all data underlying the findings in their manuscript fully available (please refer to the Data Availability Statement at the start of the manuscript PDF file)?

Reviewer #3: Yes

Reviewer #4: Yes

5. Is the manuscript presented in an intelligible fashion and written in standard English?

Reviewer #3: Yes

Reviewer #4: Yes

6. Review Comments to the Author

Reviewer #3: All the comments have been addressed. The authors have resolved all the issues raised and there are no further issues with the paper after my review.

Reviewer #4: Comments:

Title:

The manuscript uses the terms “mutuelles” and “Community-Based Health Insurance (CBHI)” interchangeably. For clarity and consistency, the authors should select one term to use throughout the paper. This will help avoid confusion for readers and ensure conceptual coherence.

Materials and Methods:

-Line 134: Please correct the typographical error.

-Line 144 (Study Period): Kindly provide a justification for the selected study period. Were there any seasonal or contextual factors that may have influenced the findings?

-Lines 153–160 (Management of the Scheme): Include details on the training and qualifications of the scheme managers. This information is important, as management professionalism can significantly impact scheme performance and outcomes.

-Line 187 (Sampling) and Line 309 (Protocol Deviation): Elaborate on the potential biases introduced by the security constraints in Mali and describe the mitigation measures taken to address these challenges.

Tables:

-Lines 342 (Table 1) and 380 (Table 2): Please improve the formatting of the tables. For example, merge the empty rows under main variable headings in the first column to enhance readability.

-Line 347: Correct the phrase “Odds gross ratios (CORs).” You may have intended to write “Crude Odds Ratios (CORs).” Please revise accordingly.

-Lines 349–368 and 381: Standardize the notation for confidence intervals. Use either “95% CI” or “CI 95%” consistently throughout the manuscript.

7. PLOS authors have the option to publish the peer review history of their article (what does this mean? ). If published, this will include your full peer review and any attached files.

**Do you want your identity to be public for this peer review?** For information about this choice, including consent withdrawal, please see our Privacy Policy .

Reviewer #3: No

Reviewer #4: No

---

## [Editor Report · Decision Letter 2]

What factors shape an individual’s probability to be enrolled in a professionally-managed Mutuelle? Results from a cross-sectional case-control study in two districts in Mali

PGPH-D-24-02388R2

Dear Dr Niangaly,

We are pleased to inform you that your manuscript 'What factors shape an individual’s probability to be enrolled in a professionally-managed Mutuelle? Results from a cross-sectional case-control study in two districts in Mali' has been provisionally accepted for publication in PLOS Global Public Health.

Best regards,

Ama Pokuaa Fenny, Ph.D

Academic Editor